# Parallelised ABox Reasoning and Query Answering with Expressive Description Logics

Andreas Steigmiller[⋆] and Birte Glimm

Ulm University, Ulm, Germany, <first name>.<last name>@uni-ulm.de

**Abstract.** Automated reasoning support is an important aspect of logic-based knowledge representation. The development of specialised procedures and sophisticated optimisation techniques significantly improved the performance even for complex reasoning tasks such as conjunctive query answering. Reasoning and query answering over knowledge bases with a large number of facts and expressive schemata remains, however, challenging.

We propose a novel approach where the reasoning over assertional knowledge is split into small, similarly sized work packages to enable a parallelised processing with tableau algorithms, which are dominantly used for reasoning with more expressive Description Logics. To retain completeness in the presence of expressive schemata, we propose a specifically designed cache that allows for controlling and synchronising the interaction between the constructed partial models. We further report on encouraging performance improvements for the implementation of the techniques in the tableau-based reasoning system Konclude.

## 1 Introduction

Description Logics (DLs) are a family of logic-based representation formalisms that provide the logical underpinning of the well-known Web Ontology Language (OWL). The knowledge expressed with DLs is typically separated into terminological (aka TBox or schema) and assertional knowledge (aka ABox or facts), where the former describes the relationships between concepts (representing sets of individuals with common characteristics) as well as roles (specifying the relationships between pairs of individuals) and the latter asserts these concepts and roles to concrete individuals of the application domain. Automated reasoning systems derive implicit consequences of the explicitly stated information, which, for example, allows for detecting modelling errors and for enriching queries by additional answers that are implied by the knowledge. Expressive DLs, such as $\mathcal{SROIQ}$ [11], allow for describing the application domain in more detail, but require sophisticated reasoning algorithms and are typically more costly in terms of computational resources. Nevertheless state-of-the-art reasoning systems are usually able to handle real-world ontologies, which often also use expressive language features, due to a large range of developed optimisation techniques.

The increasing volume of data in many application domains leads, however, also to larger amounts of assertional knowledge. For less expressive schemata (where reasoning is usually deterministic), the interest in ontology-based data access (OBDA) led

---

[⋆] Funded by the German Research Foundation (Deutsche Forschungsgemeinschaft, DFG) in project number 330492673

to several advancements, e.g., query rewriting, materialization techniques, or combined approaches. To cope with the reasoning challenges in the presence of an expressive schema several techniques have been developed, which often complement each other. There are, for example, summarisation [3,6] and abstraction techniques [8], which derive consequences for representative individuals and transfer the results to many other individuals with the same or a similar (syntactical) structure. These techniques do not necessarily work well for all ontologies, may be limited to certain queries or (fragments of) DLs, or require expensive computations, e.g., justifications. Several techniques also reduce reasoning to datalog [1,5,21] since datalog engines are targeted towards data intensive applications. This reduction, however, often leads to some additional overhead and, in some cases, it can be necessary to fall back to a fully-fledged DL reasoner, e.g., for handling non-deterministic features. Other approaches partition the ABox or extract modules out of it [20] such that each part can be processed independently [19]. Moreover, approaches based on big data principles such as map and reduce have been proposed [18]. However, they are typically also limited to specific language features and/or queries and do not work for arbitrary ontologies. Particularly challenging is the support of conjunctive queries with complex concept atoms or with existential variables that may bind to anonymous individuals, which are only implied by the knowledge base, since these features typically make it difficult to appropriately split the ABox upfront in such a way that queries can correctly be answered without too much data exchange.

Many state-of-the-art reasoners directly integrate techniques that improve ABox reasoning, e.g., (pseudo) model checking [10] or bulk processing with binary retrieval [9]. Most reasoners for expressive DLs are based on (variants of) tableau algorithms, which construct abstractions of models called completion graphs. By caching (parts of) the completion graph from the initial consistency check, subsequent reasoning tasks and queries can be answered more efficiently [12,16]. However, constructing and caching entire completion graphs for knowledge bases with large ABoxes requires significant amounts of (main) memory, which may be more than what is typically available.

In this paper, we propose to dynamically split the model construction process for tableau algorithms. This allows for (i) handling larger ABoxes since not everything has to be handled at once and for (ii) exploiting parallelisation. The proposed splits lead to similarly sized work packages that can be processed concurrently without direct synchronisation. To ensure that the partial models constructed in parallel are "compatible" with each other, we employ a cache where selected consequences for individuals are stored. For processing new or reprocessing incompatible parts of the knowledge base, we retrieve cached consequences and ensure with appropriate reuse and expansion strategies that the constructed partial models are eventually compatible with the cache, such that it can (asynchronously) be updated. Conjunctive query answering is supported by adapting the expansion criteria and by appropriately splitting the propagation work through the (partial) models for determining query answers.

The paper is organised as follows: Section 2 introduces some preliminaries about DLs and tableau algorithms; Section 3 describes the cache; Section 4 discusses the adaptations for query answering and Section 5 presents implementation details and results of experiments before we conclude in Section 6. An accompanying technical report [14] provides further details, examples, proofs, and evaluation results.

Table 1: Core features of $\mathcal{SROIQ}$ (#$M$ denotes the cardinality of the set $M$)

|  |  | Syntax | Semantics |
|---|---|---|---|
| *Individuals*: | individual | $a$ | $a^{\mathcal{I}} \in \Delta^{\mathcal{I}}$ |
| *Roles*: | atomic role | $r$ | $r^{\mathcal{I}} \subseteq \Delta^{\mathcal{I}} \times \Delta^{\mathcal{I}}$ |
|  | inverse role | $r^-$ | $\{\langle \gamma, \delta \rangle \mid \langle \delta, \gamma \rangle \in r^{\mathcal{I}}\}$ |
| *Concepts*: | atomic concept | $A$ | $A^{\mathcal{I}} \subseteq \Delta^{\mathcal{I}}$ |
|  | nominal | $\{a\}$ | $\{a^{\mathcal{I}}\}$ |
|  | top | $\top$ | $\Delta^{\mathcal{I}}$ |
|  | bottom | $\bot$ | $\emptyset$ |
|  | negation | $\neg C$ | $\Delta^{\mathcal{I}} \setminus C^{\mathcal{I}}$ |
|  | conjunction | $C \sqcap D$ | $C^{\mathcal{I}} \cap D^{\mathcal{I}}$ |
|  | disjunction | $C \sqcup D$ | $C^{\mathcal{I}} \cup D^{\mathcal{I}}$ |
|  | existential restriction | $\exists r.C$ | $\{\delta \mid \exists \gamma \in C^{\mathcal{I}} : \langle \delta, \gamma \rangle \in r^{\mathcal{I}}\}$ |
|  | universal restriction | $\forall r.C$ | $\{\delta \mid \langle \delta, \gamma \rangle \in r^{\mathcal{I}} \rightarrow \gamma \in C^{\mathcal{I}}\}$ |
|  | number restriction, $\bowtie \in \{\leqslant, \geqslant\}$ | $\bowtie n\, r.C$ | $\{\delta \mid \#\{\langle \delta, \gamma \rangle \in r^{\mathcal{I}} \text{ and } \gamma \in C^{\mathcal{I}}\} \bowtie n\}$ |
| *Axioms*: | general concept inclusion | $C \sqsubseteq D$ | $C^{\mathcal{I}} \subseteq D^{\mathcal{I}}$ |
|  | role inclusion | $r \sqsubseteq s$ | $r^{\mathcal{I}} \subseteq s^{\mathcal{I}}$ |
|  | role chains | $r_1 \circ \ldots \circ r_n \sqsubseteq S$ | $r_1^{\mathcal{I}} \circ \ldots \circ r_n^{\mathcal{I}} \subseteq S^{\mathcal{I}}$ |
| *Assertions*: | concept assertion | $C(a)$ | $a^{\mathcal{I}} \in C^{\mathcal{I}}$ |
|  | role assertion | $r(a, b)$ | $\langle a^{\mathcal{I}}, b^{\mathcal{I}} \rangle \in r^{\mathcal{I}}$ |
|  | equality assertion | $a \approx b$ | $a^{\mathcal{I}} = b^{\mathcal{I}}$ |

## 2   Description Logics and Reasoning Preliminaries

The syntax of DLs is defined using a vocabulary consisting of countably infinite pair-wise disjoint sets $N_C$ of *atomic concepts*, $N_R$ of *atomic roles*, and $N_I$ of *individuals*. A role is either atomic or an *inverse role* $r^-$, $r \in N_R$. The syntax and semantics of complex *concepts* and *axioms* are defined in Table 1. Note that we omit the presentation of some features (e.g., datatypes) and restrictions (e.g., number restrictions may not use "complex roles", i.e., roles that occur on the right-hand side of role chains or are implied by such roles) for brevity. A knowledge base/ontology $\mathcal{K}$ is a finite set of axioms. One typically distinguishes terminological axioms in the TBox $\mathcal{T}$ (e.g., $C \sqsubseteq D$) and assertions in the ABox $\mathcal{A}$ (e.g., $C(a)$) of $\mathcal{K}$, i.e., $\mathcal{K} = (\mathcal{T}, \mathcal{A})$. We use $\mathsf{inds}(\mathcal{K})$ to refer to the individuals of $\mathcal{K}$. An *interpretation* $\mathcal{I} = (\Delta^{\mathcal{I}}, \cdot^{\mathcal{I}})$ consists of a non-empty *domain* $\Delta^{\mathcal{I}}$ and an *interpretation function* $\cdot^{\mathcal{I}}$. We say that $\mathcal{I}$ *satisfies* a general concept inclusion (GCI) $C \sqsubseteq D$, written $\mathcal{I} \models C \sqsubseteq D$, if $C^{\mathcal{I}} \subseteq D^{\mathcal{I}}$ (analogously for other axioms and assertions as shown in Table 1). If $\mathcal{I}$ satisfies all axioms of $\mathcal{K}$, $\mathcal{I}$ is a *model* of $\mathcal{K}$ and $\mathcal{K}$ is *consistent/satisfiable* if it has a model.

A tableau algorithm decides the consistency of a knowledge base $\mathcal{K}$ by trying to construct an abstraction of a model for $\mathcal{K}$, a so-called *completion graph*. A completion graph $G$ is a tuple $(V, E, \mathcal{L}, \dot{\neq})$, where each node $v \in V$ (edge $\langle v, w \rangle \in E$) represents one or more (pairs of) individuals. Each node $v$ (edge $\langle v, w \rangle$) is labelled with a set of concepts (roles), $\mathcal{L}(v)$ ($\mathcal{L}(\langle v, w \rangle)$), which the individuals represented by $v$ ($\langle v, w \rangle$) are instances of. The relation $\dot{\neq}$ records inequalities between nodes. We call $C \in \mathcal{L}(v)$ ($r \in \mathcal{L}(\langle v, w \rangle)$) a *concept (role) fact*, which we write as $C(v)$ ($r(v, w)$). A node $v$ is a *nominal node* if $\{a\} \in \mathcal{L}(v)$ for some individual $a$ and a *blockable node* otherwise. For

$r \in \mathcal{L}(\langle v, w \rangle)$ and $r \sqsubseteq^* s$, with $\sqsubseteq^*$ the reflexive, transitive closure over role inclusions of $\mathcal{K}$ (including their inverses), we call $w$ an *s-successor* of $v$ and $v$ an *s-predecessor* of $w$. A node $w$ is called an *s-neighbour* of $v$ if $w$ is an $s$-successor of $v$ or $v$ an $s^-$-*successor* of $w$. We use *ancestor* and *descendant* as the transitive closure of the predecessor and successor relation, respectively. We say that $v_n$ is an *implied descendant* of $v_0$ if there is a path $v_0, v_1, \ldots, v_n$ such that $v_{i+1}$ is a successor of $v_i$ for $0 \leq i < n$ and each $v_j$ with $j > 0$ does not represent an individual of $\mathsf{inds}(\mathcal{K})$.

A completion graph is initialised with one node for each individual in the input knowledge base. Concepts and roles are added to the node and edge labels as specified by concept and role assertions. For simplicity, we assume that, for each individual $a \in \mathsf{inds}(\mathcal{K})$, a nominal $\{a\}$ is added to $\mathcal{L}(v_a)$. This allows for easily handling equality assertions by adding $\{b\}$ to $\mathcal{L}(v_a)$ for $a \approx b \in \mathcal{A}$. As a convention, we write $v_a$ to refer to the node representing $a \in \mathsf{inds}(\mathcal{K})$, i.e., $\{a\} \in \mathcal{L}(v_a)$. Note that $v_a$ and $v_b$ can refer to the same node if $\{a\}$ and $\{b\}$ are in its label. Complex concepts are then decomposed using a set of expansion rules, where each rule application can add new concepts to node labels and/or new nodes and edges, thereby explicating the structure of a model. The rules are applied until either the graph is *fully expanded* (no more rules are applicable), in which case the graph can be used to construct a model that is a *witness* to the consistency of $\mathcal{K}$, or an obvious contradiction (called a *clash*) is discovered (e.g., a node $v$ with $C, \neg C \in \mathcal{L}(v)$), proving that the completion graph does not correspond to a model. $\mathcal{K}$ is consistent if the rules (some of which are non-deterministic) can be applied such that they build a fully expanded, clash-free completion graph. Cycle detection techniques such as *pairwise blocking* [11] prevent the infinite generation of new nodes.

## 3   Caching Individual Derivations

Since tableau-based reasoning algorithms reduce (most) reasoning tasks to consistency checking, parallelising the completion graph construction has general benefits on the now ubiquitous multi-core processor systems. Partitioning the ABox upfront such that no or little interaction is required between the partitions [19] no longer works for expressive DLs, such as $\mathcal{SROIQ}$, or complex reasoning tasks, such as conjunctive query answering (with complex concepts and/or existential variables). This is, for example, due to implied connections between individuals (e.g., due to nominals) or due to the consideration of new concept expressions at query time. The effect of parallelisation is further hindered by the multitude of optimisations, required to properly deal with real-world ontologies, which often introduce dependencies between rules and (parts of) completion graphs, resulting in the need of data synchronisation. For example, the anywhere blocking optimisation (cycle detection) investigates all previously constructed nodes in the completion graph in order to determine whether a node is blocked. Hence, a parallelisation approach where a completion graph is modified in parallel can be difficult to realise since it could require a lot of synchronisation.

For ontologies with large ABoxes, it seems more suitable to build completion graphs for parts of the ABox separately (by independent threads) and, since independence of the parts cannot be assumed, to align the results afterwards. Such an alignment can, however, be non-trivial on several levels: For example, if different non-deterministic

decisions have been made for individuals in overlapping parts or due to technical details of the often complex data structures for completion graphs, e.g., efficient processing queues, caching status of node labels, etc.

Our parallelisation approach focuses on aligning completion graphs for ABox parts and we address the challenges by employing a cache for certain derivations for individuals, which facilitates the alignment process. For this, consistency checking roughly proceeds as follows: We randomly split the ABox into equally sized parts that are distributed to worker threads. When a thread begins to process one of these ABox parts, it retrieves stored derivations from the cache for (possibly) affected individuals in that part. The thread then tries to construct a fully expanded and clash-free *local* completion graph for the ABox part by reusing cached derivations and/or by expanding the processing to individuals until they are "compatible" with the cache. Compatibility requires that the local completion graph is fully expanded as well as clash-free and that it can be expanded such that it matches the derivations for the remaining individuals in the cache. If it is required to extend the processing to some "neighbouring" individuals for achieving compatibility (e.g., if different non-deterministic decisions are required for the already processed individuals), then also the cached derivations for these individuals are retrieved and considered. If this process succeeds, the cache is updated with the new or changed derivations for the processed individuals.

If compatibility cannot be obtained (e.g., due to expansion limitations that ensure similarly sized work packages), then the cache entries of incompletely handled individuals are marked such that they are considered later separately. For this, a thread loads the data for (some) marked individuals and tries to construct a fully expanded and clash-free completion graph for them until full compatibility is obtained. If clashes occur that depend on reused (non-deterministic) derivations from the cache, then the corresponding individuals can be identified such that their expansion can be prioritized and/or the reuse of their derivations can be avoided. As a result, (in)consistency of the knowledge base can eventually be detected, as soon as all problematic individuals are directly expanded and all relevant non-deterministic decisions are investigated together.

Before describing the different aspects of the approach and the work-flow in more detail, we define a basic version of the cache and how derivations are stored and used.

**Definition 1 (Individual Derivations Cache).** *Let $\mathcal{K}$ be a knowledge base. We use* $\mathsf{fclos}(\mathcal{K})$*,* $\mathsf{rols}(\mathcal{K})$*, and* $\mathsf{inds}(\mathcal{K})$ *for the sets of concepts, roles, and individuals that can occur in $\mathcal{K}$ or in a completion graph for $\mathcal{K}$. An* individual derivations cache $C$ *is a (partial) mapping of individuals from* $\mathsf{inds}(\mathcal{K})$ *to* cache entries*, where the cache entry for an individual $a \in \mathsf{inds}(\mathcal{K})$ consists of:*

- $K^C \subseteq 2^{\mathsf{fclos}(\mathcal{K})}$ *and* $P^C \subseteq 2^{\mathsf{fclos}(\mathcal{K})}$*: the sets of known and possibly instantiated concepts of a, respectively,*
- $I \subseteq 2^{\mathsf{inds}(\mathcal{K})}$*: the individuals that are (indirectly) connected via nominals to a,*
- $\exists\colon \mathsf{rols}(\mathcal{K}) \to \mathbf{N}_0$*: mapping a role r to the number of existentially derived successors for a and r, and*
- $K^R\colon \mathsf{rols}(\mathcal{K}) \to 2^{\mathsf{inds}(\mathcal{K})}$ *and* $P^R\colon \mathsf{rols}(\mathcal{K}) \to 2^{\mathsf{inds}(\mathcal{K})}$*: mapping a role r to the sets of known and possible neighbours of a and r, respectively.*

*We write $K^C(a, C)$, $P^C(a, C)$, $I(a, C)$, $\exists(a, C)$, $K^R(a, C)$, and $P^R(a, C)$ to refer to the individual parts of the cache entry $C(a)$. We write $a \in C$ if $C$ is defined for a.*

Note that we distinguish between known and possible information in the cache, which mostly correspond to the deterministically and non-deterministically derived consequences in completion graphs. Non-deterministically derived facts in completion graphs can usually be identified via branching tags for dependency directed backtracking [2,17]. If the cache is clear from the context, we simply write $K^C(a)$ or $K^R(a)(r)$, where the latter returns the known (deterministically derived) $r$-neighbours of $a$.

 Let $\mathcal{K} = (\mathcal{T}, \mathcal{A})$ be a knowledge base and $\mathcal{A}_j \subseteq \mathcal{A}$ the processed ABox part. In addition to the usual initialisation of a completion graph $G = (V, E, \mathcal{L}, \dot{\neq})$ for $\mathcal{K} = (\mathcal{T}, \mathcal{A}_j)$, we add $K^C(a)$ to $\mathcal{L}(v_a)$ and $r$ to $\mathcal{L}(\langle v_a, v_b \rangle)$ if $b \in K^R(a)(r)$, for each $v_a, v_b \in V$. If a node $v \in V$ exists with $\{c\} \in \mathcal{L}(v)$ or $\neg\{c\} \in \mathcal{L}(v)$, but $v_c \notin V$, then we add $v_c$ with $\{c\} \in \mathcal{L}(v_c)$ to $V$ and initialise $v_c$ analogously. Once $G$ is extended into a fully expanded and clash-free completion graph, we identify the derivations for cache entries for each individual $a$ with $v_a \in V$. Since deterministically derived consequences remain valid for all possible completion graphs, we update the corresponding cache entries by adding the deterministically derived consequences, e.g., for $K^C$, whereas non-deterministically derived consequences may change and, thus, replace existing cache entries, e.g., for $P^C$. A completion graph for $(\mathcal{T}, \mathcal{A}_j)$ is *compatible* with the cache if it can be extended to a fully expanded and clash-free completion graph for $(\mathcal{T}, \mathcal{A}_1 \cup \ldots \cup \mathcal{A}_j)$, where $\mathcal{A}_1 \cup \ldots \cup \mathcal{A}_{j-1}$ are the previously processed (and cached) ABox parts. As argued above, this might require the integration and processing of individuals from the cache during the completion graph expansion. Hence, we define when individuals in the cache *potentially influence* or are *influenced by* the completion graph.

**Definition 2 (Cache Influence and Compatibility).** *Let $\mathcal{K} = (\mathcal{T}, \mathcal{A})$ be a knowledge base, $G = (V, E, \mathcal{L}, \dot{\neq})$ a completion graph for $(\mathcal{T}, \mathcal{A}_j)$ with $\mathcal{A}_j \subseteq \mathcal{A}$, and $v_a \in V$. We use $\#\mathsf{exrols}_r(v_a)$ for the number of $r$-neighbours of $v_a$ that do not represent an individual of $\mathsf{inds}(\mathcal{K})$ and $\mathsf{neighb}_r(v_a)$ for the $r$-neighbours of $v_a$ that represent an individual of $\mathsf{inds}(\mathcal{K})$. For an individual derivations cache $C$ (c.f. Def. 1), an individual $b \in C$ with $v_b \notin V$ is* potentially influenced *by $G$ if*

D1 $\forall r.C \in \mathcal{L}(v_a), b \in K^R(a)(r) \cup P^R(a)(r)$, and $C \notin K^C(b) \cup P^C(b)$;
D2 $\leqslant n\, r.C \in \mathcal{L}(v_a)$, $b \in K^R(a)(r) \cup P^R(a)(r)$, and $\{C, \neg C\} \cap (K^C(b) \cup P^C(b)) = \emptyset$;
D3 $\leqslant n\, r.C \in \mathcal{L}(v_a)$, $b \in K^R(a)(r) \cup P^R(a)(r)$, and $\#[\{d \mid d \in K^R(c)(r) \cup P^R(c)(r)$ with $\{c\} \in \mathcal{L}(v_a)\} \cup \mathsf{neighb}_r(v_a)] + \#\mathsf{exrols}_r(v_a) > n$;
D4 $b \in I(a)$ and $C \in \mathcal{L}(v_a)$, $C \notin K^C(a) \cup P^C(a)$ or $\leqslant n\, r.C \in \mathcal{L}(v_a)$ with $\exists(a)(r) > 0$; or
D5 $\{c\} \in \mathcal{L}(v_a), \{c\} \notin K^C(a) \cup P^C(a)$, $a \in K^R(b)(r) \cup P^R(b)(r)$ and $c \notin K^R(b)(r) \cup P^R(b)(r)$ or $b \in K^R(a)(s) \cup P^R(a)(s)$ and $b \notin K^R(c)(s) \cup P^R(c)(s)$.

*An individual $b \in C$ with $v_b \notin V$ is* potentially influencing *the completion graph $G$ if*

G1 $b \in P^R(a)(r)$;       G3 $b \in I(a)$, $C \in P^C(a)$, $C \notin \mathcal{L}(v_a)$; or
G2 $b \in K^R(a)(r)$, $C \in P^C(a)$, $C \notin \mathcal{L}(v_a)$;  G4 $\{a\} \in P^C(b)$ or $\neg\{a\} \in P^C(b)$;

*We say that $G$ is* compatible *with a cache $C$ if there is no individual $b$ that is potentially influenced by $G$ or potentially influencing $G$.*

 Roughly speaking, an individual is potentially influenced by a completion graph if integrating it into the completion graph could lead to new consequences being propagated to it. In contrast, an individual potentially influences a completion graph if the integration of it could result in new consequences for the local completion graph.

The conditions for determining influenced individuals have a strong correspondence with the tableau expansion rules (cf. [11]). In fact, Condition D1, D2, and D3 basically check whether the $\forall$-, the choose-, or the $\leqslant$-rule could potentially be applicable, i.e., whether the handling of $\forall r.C$ or $\leqslant r\,n.C$ concepts requires the consideration of neighbours from the cache. The first part of Condition D4 further checks whether some individual is indirectly connected to $a$ via a nominal for an implied descendant for an individual $b$ and whether new consequences could be propagated to that descendant. This could be the case if the label for $a$ differs to the concepts for $a$ in the cache. The second part handles potential cases where new nominals may have to be introduced and may influence $b$ or descendants of $b$. Finally, Condition D5 ensures that neighbours are integrated if individuals are newly merged such that their neighbour relations in the cache can be updated.

Instead of mirroring Conditions D1–D4 for determining influencing individuals, we use the relatively simple Conditions G1–G4 since they allow for simpler data structures and efficient cache updates. Condition G1 and G2 simply check whether some possible instances are missing in the local completion graph, which may stem from a neighbouring individual from the cache. Condition G3 analogously checks for a potentially influencing individual $b$ that is indirectly connected via the nominal $\{a\}$ in the label of an implied descendant of $b$. Condition G4 checks for merges and inequality information caused by non-deterministically derived nominal expressions for other individuals.

The following example, inspired by the well-known UOBM ontology, illustrates consistency checking with the cache.

*Example 1.* Suppose an ABox consisting of the two parts:

$\mathcal{A}_1 = \{\; \forall enr^-.(\forall takes.GC \sqcup \forall takes.UGC)(uni), \quad likes(stud, soccer), \quad enr(stud, uni),$
$\qquad\quad \forall likes^-.SoccerFan(soccer), \qquad\qquad takes(stud, course)\},$

$\mathcal{A}_2 = \{\; \exists hc.\exists likes.\{soccer\}(prof), \qquad\qquad teaches(prof, course),$
$\qquad\quad \forall teaches.\forall takes^-.\neg TennisFan(prof), \quad likes(prof, soccer)\}.$

We abbreviate *Undergraduate Course* as *UGC*, *Graduate Course* as *GC*, *enrolled in* as *enr*, *has child* as *hc*, and *student* as *stud*. For checking $\mathcal{A}_1$, we initialise a completion graph with nodes and edges that reflect the individuals and assertions in $\mathcal{A}_1$ (cf. upper part of Figure 1). To satisfy the universal restriction for $v_{soccer}$, we apply the $\forall$-rule, which propagates *SoccerFan* to $v_{stud}$. Analogously, the universal restriction for $v_{uni}$ propagates $\forall takes.GC \sqcup \forall takes.UGC$ to $v_{stud}$. We assume that the disjunct $\forall takes.GC$ is checked first, i.e., it is non-deterministically added to $\mathcal{L}(v_{stud})$. Then the concept *GC* is propagated to $v_{course}$. The completion graph for $\mathcal{A}_1$ is now fully expanded and clash-free. We next extract the data for the cache (as shown in the lower part of Figure 1).

The completion graph for $\mathcal{A}_2$ is analogously initialised (cf. upper part of Figure 2). For the concept $\exists hc.\exists likes.\{soccer\} \in \mathcal{L}(v_{prof})$, the $\exists$-rule of the tableau algorithm builds a blockable *hc*-successor for $v_{prof}$ with $\exists likes.\{soccer\}$ in its label, for which another successor is created that is merged with $v_{soccer}$ (due to the nominal) leading to the depicted edge to $v_{soccer}$. Due to the universal restriction $\forall likes^-.SoccerFan$ in $\mathcal{L}(v_{soccer})$, *SoccerFan* is propagated to $v_1$ and to $v_{prof}$. For the universal restriction $\forall teaches.\forall takes^-.\neg TennisFan \in \mathcal{L}(v_{prof})$, we propagate $\forall takes^-.\neg TennisFan$ to $v_{course}$. Now, there are no more tableau expansion rules applicable to the constructed com-

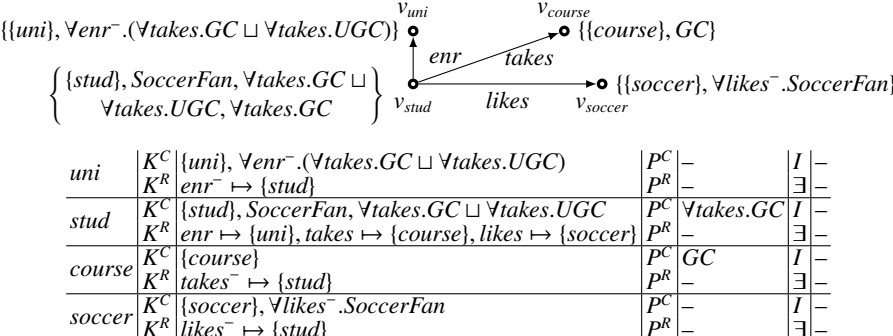

| | | $K^C$ | $\{uni\}, \forall enr^-.(\forall takes.GC \sqcup \forall takes.UGC)$ | $P^C$ | $-$ | $I$ | $-$ |
|---|---|---|---|---|---|---|---|
| $uni$ | | $K^R$ | $enr^- \mapsto \{stud\}$ | $P^R$ | $-$ | $\exists$ | $-$ |
| $stud$ | | $K^C$ | $\{stud\}, SoccerFan, \forall takes.GC \sqcup \forall takes.UGC$ | $P^C$ | $\forall takes.GC$ | $I$ | $-$ |
| | | $K^R$ | $enr \mapsto \{uni\}, takes \mapsto \{course\}, likes \mapsto \{soccer\}$ | $P^R$ | $-$ | $\exists$ | $-$ |
| $course$ | | $K^C$ | $\{course\}$ | $P^C$ | $GC$ | $I$ | $-$ |
| | | $K^R$ | $takes^- \mapsto \{stud\}$ | $P^R$ | $-$ | $\exists$ | $-$ |
| $soccer$ | | $K^C$ | $\{soccer\}, \forall likes^-.SoccerFan$ | $P^C$ | $-$ | $I$ | $-$ |
| | | $K^R$ | $likes^- \mapsto \{stud\}$ | $P^R$ | $-$ | $\exists$ | $-$ |

Fig. 1: Local completion graph (upper part) and entries of the individual derivations cache (lower part) for handling ABox $\mathcal{A}_1$ of Example 1

pletion graph, but it is not yet compatible with the cache and we have to integrate (potentially) influenced or influencing individuals. In fact, *course* causes two incompatibilities: On the one hand, Condition D1 identifies *stud* as (potentially) influenced due to $\forall takes^-.\neg TennisFan \in \mathcal{L}(v_{course})$ and because *stud* is a $takes^-$-neighbour of *course* according to the cache (cf. Figure 1). On the other hand, Condition G2 is satisfied (since $GC \notin \mathcal{L}(v_{course})$ but $GC \in P^C(course)$) and, therefore, the neighbour *stud* listed in $K^R(course)(takes^-)$ is identified as potentially influencing. We integrate *stud* by creating the node $v_{stud}$, by adding the concepts $\{stud\}$ and $\forall takes.GC \sqcup \forall takes.UGC$ from the cache to $\mathcal{L}(v_{stud})$, and by creating an edge to $v_{course}$ labelled with *takes* as well as an edge to $v_{soccer}$ labelled with *likes*. Now, the rule application for $\forall takes^-.\neg TennisFan \in \mathcal{L}(course)$ propagates $\neg TennisFan$ to $v_{stud}$. In addition, by reprocessing the disjunction $\forall takes.GC \sqcup \forall takes.UGC$ for $v_{stud}$, we obtain $GC \in \mathcal{L}(v_{course})$ if the same disjunct is chosen. As a result, the completion graph is fully expanded and clash-free w.r.t. $\mathcal{A}_2$ and it is compatible with the cache. Hence, the cache can be updated resulting in the entries depicted in the lower part of Figure 2. Note that only $v_{uni}$ has not been integrated in the completion graph for $\mathcal{A}_2$, but there is usually a bigger gain for larger ABoxes.

For parallelising the work-flow with the cache, one has to update and use the entries in a consistent/atomic way such that the state is clear. This can efficiently (and asynchronously) be realised by associating an update id with each cached individual and allow them to have an "inconsistent state" in the cache (see technical report for details). If an update extracted from a constructed completion graph refers to a cache entry of an individual with a changed update id (i.e., the cache entry has been changed by another thread), then the non-deterministically derived consequences are simply added and the state of the individual is set inconsistent (e.g., by adding $\bot$ to $P^C(a)$). If all parts of the ABox are processed, then we repeatedly retrieve individuals with inconsistent states from the cache and reprocess them until compatibility is achieved. Repeatedly deriving different consequences for individuals and correspondingly updating the cache in parallel can, however, threaten termination of the procedure. To retain termination, we simply increase the number of individuals that are processed by one thread such that, in the worst-case, all individuals with inconsistent states are eventually processed together.

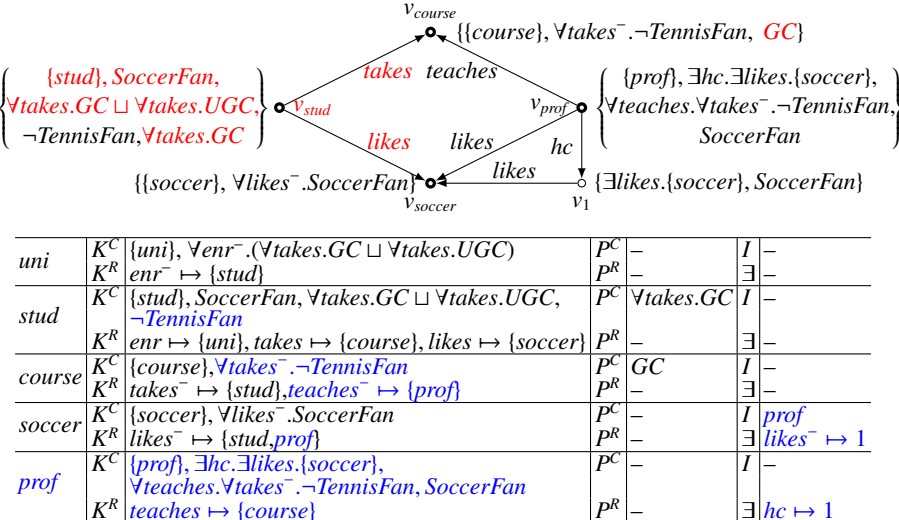

Fig. 2: Local completion graph (upper part, expansions from cache due to incompatibilities in red) and entries of the individual derivations cache (lower part, changes in blue) for handling ABox $\mathcal{A}_2$ of Example 1

A naive expansion to all influenced individuals can cause a significant work imbalance for knowledge bases that use complex roles or have intensively connected individuals. In fact, if many neighbour individuals are influenced by universal restrictions or by the merging of nodes, then the expansion to all of them could result in an enormous completion graph processed by one thread, which could limit the effectiveness of the parallelisation. This can be addressed by "cutting the propagation" in the completion graph, for which we then notify the cache that the states of the remaining individuals have to be considered inconsistent such that their processing can be continued later. This cannot avoid the (theoretical) worst-case of processing the whole knowledge base in the end, but it seems to work well for many real-world ontologies, as indicated by our evaluation.

## 4  Query Answering Support

Compared to other more sophisticated reasoning tasks, conjunctive query answering is typically more challenging since an efficient reduction to consistency checking is not easy. However, a new approach for answering (conjunctive) queries has recently been introduced, where the query atoms are "absorbed" into several simple DL-axioms [13]. These "query axioms" are of the form $C \sqsubseteq \downarrow x.S^x$, $S^x \sqsubseteq \forall r.S^x_r$, $S^x \sqcap A \sqsubseteq S^x_A$, and $S^x \sqcap S^y \sqsubseteq S^{xy}$, where $\downarrow x.S^x$ is a binder concept that triggers the creation of variable mappings in the extended tableau algorithm and $S$ (possibly with sub- and/or superscripts) are so-called query state concepts that are associated with variable mappings, as defined in the following, in order to keep track of partial matches of the query in a completion graph.

**Definition 3 (Variable Mappings).** *A* variable mapping $\mu$ *is a (partial) function from variable names to nodes. Let* $G = (V, E, \mathcal{L}, \neq, \mathcal{M})$ *be an (extended) completion graph, where* $\mathcal{M}(C, v)$ *denotes the sets of variable mappings that are associated with a concept* $C$ *in* $\mathcal{L}(v)$. *A variable mapping* $\mu_1 \cup \mu_2$ *is defined by setting* $(\mu_1 \cup \mu_2)(x) = \mu_1(x)$ *if* $x$ *is in the domain of* $\mu_1$, *and* $(\mu_1 \cup \mu_2)(x) = \mu_2(x)$ *otherwise. Two variable mappings* $\mu_1$ *and* $\mu_2$ *are* compatible *if* $\mu_1(x) = \mu_2(x)$ *for all* $x$ *in the domain of* $\mu_1$ *as well as* $\mu_2$. *The* join $\mathcal{M}_1 \bowtie \mathcal{M}_2$ *between the sets of variable mappings* $\mathcal{M}_1$ *and* $\mathcal{M}_2$ *is defined as:*

$$\mathcal{M}_1 \bowtie \mathcal{M}_2 = \{\mu_1 \cup \mu_2 \mid \mu_1 \in \mathcal{M}_1, \mu_2 \in \mathcal{M}_2 \text{ and } \mu_1 \text{ is compatible with } \mu_2\}.$$

Rules of the extended tableau algorithm are shown in Table 2 (without considering blocking), which handle the new axioms and concepts by correspondingly creating and propagating variable mappings. For example, a binder concept $\downarrow x.S^x \in \mathcal{L}(v)$ is handled by adding $S^x$ to $\mathcal{L}(v)$ and by creating a mapping $\{x \mapsto v\}$ that is associated with $S^x$ for $v$, i.e., $\{x \mapsto v\} \in \mathcal{M}(S^x, v)$. Although conjunctive query answering with arbitrary existential variables is still open for $\mathcal{SROIQ}$, the approach works for knowledge bases where only a limited number of new nominal nodes is enforced (by using an extended analogous propagation blocking technique) [13], which is generally the case in practice.

As an example, a simple query with only the atoms $r(x, y)$ and $s(y, x)$ (with $x, y$ both answer variables) can systematically be absorbed into the axioms $\top \sqsubseteq \downarrow x.S^x$, $S^x \sqsubseteq \forall r.S_r^x$, $S_r^x \sqsubseteq \downarrow y.S^y$, $S_r^x \sqcap S^y \sqsubseteq S^{xy}$, $S^{xy} \sqsubseteq \forall s.S_s^{xy}$, and $S_s^{xy} \sqcap S^x \sqsubseteq S^{xyx}$. The query state concept $S_r^x$, for example, represents the state where bindings for $x$ are propagated to $r$-successors, i.e., $r(x, y)$ is satisfied. For bindings that are propagated back over $s$-edges via $\forall s.S_s^{xy}$, the final binary inclusion axiom checks whether the cycle is closed. If it is, the joined variable mappings are associated with $S^{xyx}$ from which answer candidates can be extracted once a fully expanded and clash-free completion graph is found.

Note that with sophisticated absorption techniques, variable mappings can often be derived deterministically, i.e., they directly constitute query answers. Non-deterministically obtained variable mappings do, however, require a separate entailment check to verify that there exist no counter example with the query variables equally bound as in the non-deterministically derived variable mapping. This can be realised by restricting the generated binder concepts of the absorption process to only create corresponding bindings and by triggering a clash with the additional axiom $S^{xyx} \sqsubseteq \bot$.

While query answering by absorption is able to process queries for many (expressive) real-world ontologies [13], especially queries with existential variables can require a substantial amount of computation. A significant bottleneck is often the (variable mappings) *propagation task*, i.e., the creation and propagation of the variable mappings to get all potential answers from a completion graph. Building and using completion graphs for partial ABoxes (possibly in parallel) is difficult since it is unclear which joins of bindings can occur in answers and, hence, how the ABox can suitably be partitioned. The individual derivations cache can, however, also help to split the propagation work for variable mappings such that each thread can completely determine a few answer candidates over a part of the ABox. This enables (a uniform) parallelisation of the propagation task. Note that a partitioning of individuals for the first variable can be used for a naive parallelisation: We create several propagation tasks and restrict the $\downarrow$-rule to bind the first variable only to the individuals of the handled partition. This can, however, lead to a work imbalance if many answers are based on the same individual for the first

Table 2: Tableau rule extensions

| $\downarrow$-rule: |
|---|
| if $\quad \downarrow x.C \in \mathcal{L}(v)$, and $C \notin \mathcal{L}(v)$ or $\{x \mapsto v\} \notin \mathcal{M}(C, v)$ |
| then $\mathcal{L}(v) = \mathcal{L}(v) \cup \{C\}$, $\mathcal{M}(C, v) = \mathcal{M}(C, v) \cup \{\{x \mapsto v\}\}$ |
| $\forall$-rule: |
| if $\quad \forall r.C \in \mathcal{L}(v)$, there is an $r$-neighbour $w$ of $v$ with $C \notin \mathcal{L}(w)$ or $\mathcal{M}(\forall r.C, v) \nsubseteq \mathcal{M}(C, w)$ |
| then $\mathcal{L}(w) = \mathcal{L}(w) \cup \{C\}$, $\mathcal{M}(C, w) = \mathcal{M}(C, w) \cup \mathcal{M}(\forall r.C, v)$ |
| $\sqsubseteq_1$-rule: |
| if $\quad S^{x_1 \cdots x_n} \sqsubseteq C \in \mathcal{K}$, $S^{x_1 \cdots x_n} \in \mathcal{L}(v)$, and $C \notin \mathcal{L}(v)$ or $\mathcal{M}(S^{x_1 \cdots x_n}, v) \nsubseteq \mathcal{M}(C, v)$ |
| then $\mathcal{L}(v) = \mathcal{L}(v) \cup \{C\}$, $\mathcal{M}(C, v) = \mathcal{M}(C, v) \cup \mathcal{M}(S^{x_1 \cdots x_n}, v)$ |
| $\sqsubseteq_2$-rule: |
| if $\quad S^{x_1 \cdots x_n} \sqcap A \sqsubseteq C \in \mathcal{K}$, $\{S^{x_1 \cdots x_n}, A\} \subseteq \mathcal{L}(v)$, and $C \notin \mathcal{L}(v)$ or $\mathcal{M}(S^{x_1 \cdots x_n}, v) \nsubseteq \mathcal{M}(C, v)$ |
| then $\mathcal{L}(v) = \mathcal{L}(v) \cup \{C\}$, $\mathcal{M}(C, v) = \mathcal{M}(C, v) \cup \mathcal{M}(S^{x_1 \cdots x_n}, v)$ |
| $\sqsubseteq_3$-rule: |
| if $\quad S_1^{x_1 \cdots x_n} \sqcap S_2^{y_1 \cdots y_m} \sqsubseteq C \in \mathcal{K}$, $\{S_1^{x_1 \cdots x_n}, S_2^{y_1 \cdots y_m}\} \subseteq \mathcal{L}(v)$, and $(\mathcal{M}(S_1^{x_1 \cdots x_n}, v) \bowtie \mathcal{M}(S_2^{y_1 \cdots y_m}, v)) \nsubseteq \mathcal{M}(C, v)$ |
| then $\mathcal{L}(v) = \mathcal{L}(v) \cup \{C\}$, $\mathcal{M}(C, v) = \mathcal{M}(C, v) \cup (\mathcal{M}(S_1^{x_1 \cdots x_n}, v) \bowtie \mathcal{M}(S_2^{y_1 \cdots y_m}, v))$ |

---

**Algorithm 1** recPropTask$(R, i)$

**Input:** Variable binding restrictions $R$ and the index of the next to be handled variable

```
1:  if i ≤ n then
2:      B ← recPropTask(R, i + 1)
3:      for each xⱼ with 1 ≤ j < i do
4:          R(xⱼ) ← B(xⱼ)
5:      end for
6:      while |B(xᵢ)| ≥ l do
7:          R(xᵢ) ← R(xᵢ) \ B(xᵢ)
8:          Bₜ ← recPropTask(R, i + 1)
9:          B(xᵢ) ← Bₜ(xᵢ)
10:     end while
11:     B(xᵢ) ← ∅
12: else
13:     G ← buildComplGraph(R, l)
14:     C ← C ∪ answerCands(G)
15:     B ← extractBoundIndis(G)
16: end if
17: return B            ▷ Returning
    the bound individuals from the last
    constructed completion graph
```

variable and we cannot easily impose restrictions for the other variables since we do not know which combinations of individuals can occur in answers.

We can, however, use a dynamic approach, where we limit the number of individuals to which a variable can be bound. Each individual bound to such a "binding-limited" variable is recorded and in the next propagation task we exclude bindings to already tested individuals. This can, for example, be realised with the recursive function recPropTask sketched in Algorithm 1, which takes as input a mapping $R$ from variables to (still) allowed bindings for individuals and the index $i$ of the current variable (assuming that the variables are sorted in the order in which they are absorbed). The function accesses and modifies some variables via side effects, namely $l$, denoting the limit for the number of allowed bindings for each variable, $n$, standing for the number of variables in the query, and $C$, denoting the set of answer candidates. The function is initially called with $R(x) = \mathsf{inds}(\mathcal{K})$ for each variable $x$ in the query and with $i = 1$ such that the restrictions for the first variable are managed first. As long as there are more variables to handle, the function calls itself recursively for the next variable (cf. Line 2) and checks for the returned sets of bound individuals, denoted with $B$, from the last generated completion graph whether the limit $l$ has been reached for the current variable. If this is the case, then the bindings for previous variables are "frozen", i.e., they are interpreted as the only allowed bindings (cf. Line 3–5), and the used bindings for the current variable are excluded for the next propagation task (cf. Line 7). This ensures that all combina-

tions are tested step-by-step and that each propagation task only creates and propagates a limited amount of variable mappings. In fact, if the restrictions for all variables are set, then they are used for constructing the next completion graph (Line 13), where $R$ and $l$ are checked by an adapted $\downarrow$-rule. Subsequently, we can extract the found answer candidates (Line 14) and the individuals that have been used for bindings (Line 15).

Note that the concepts from the query absorption typically cause an expansion of the local completion graph due to influence criteria, e.g., if $\forall r.S_r^x$ is in the label of some node, then Condition G1 identifies all $r$-neighbours from the cache as (potentially) influenced and the corresponding nodes need to be integrated into the completion graph to propagate the associated variable mappings to them. This can result in significant propagation work, in particular, for complex roles and individuals with many neighbours. Moreover, exhausted binding restrictions for the next variable might prevent us from actually using the mappings. To address this, one can impose propagation restrictions for universal restrictions of the form $\forall r.S_r^x$ such that the local completion graph is only expanded to nodes for which bindings are possible for the next variable. This can easily be implemented by adapting the query absorption to annotate universal restrictions with the variable of the role atom to which the propagation occurs. For example, a concept of the form $\forall r.S_r^x$ resulting from the query atom $r(x, y)$ is annotated with $y$, denoted as $\forall r_{\rightarrow y}.S_r^x$. Condition D1 is then adapted to only identify individuals as influenced that are allowed as bindings for the labelled variable of the universal restriction.

**Definition 4 (Query Propagation Influence).** *Let $G = (V, E, \mathcal{L}, \dot{\neq}, \mathcal{M})$ be an (extended) completion graph for a knowledge base $\mathcal{K}$, C an individual derivations cache (c.f. Def. 1), $v_a \in V$ a node representing the individual a, and y a query variable. A restriction set $R(y) \subseteq \mathsf{inds}(\mathcal{K})$ for y restricts the individuals to which y can be bound, i.e., only to a node $v_a$ if $a \in R(y)$. An individual $b \in C$ such that no node in V contains $\{b\}$ in its label is* (query propagation) influenced *if*

Q1  $\forall r_{\rightarrow y}.S_r^{x_1,\ldots,x_n} \in \mathcal{L}(v_a)$, $b \in K^R(a)(r) \cup P^R(a)(r)$, and $b \in R(y)$.

As mentioned, if the restrictions for a variable are not known upfront, then one can collect them dynamically by only imposing a limit for the number of individuals for the restriction set. While we check whether an individual $b$ is query propagation influenced w.r.t. a variable $y$ and the amount of individuals in the restriction set $R(y)$ is less than the limit, we simply add $b$ to $R(y)$ such that Condition Q1 is satisfied. Analogously, we add $b$ to $R(y)$ when we test whether we can bind $y$ to $b$ for $\downarrow$-concepts and the limit is not yet reached. When the limit is reached, no more individuals are added to the restriction set and, therefore, no other (combinations of) variable mappings are created and the completion graph is not further expanded to other individuals. The collected restrictions are then used in the next propagation task to enforce the exploration of other (combinations of) variable mappings. Note, however, that steering the expansion with the query propagation influence condition cannot straightforwardly be used for roles with recursive role inclusion axioms (e.g., transitive roles) due to the unfolding process and since it would be too restrictive. One could possibly improve the handling for complex roles with non-trivial adaptations to the tableau algorithm, but it is unclear whether this is worth the effort. In particular, even if the individuals are expanded, the binding restrictions are already adhered to by the $\downarrow$-rule and one can simply prioritise the absorption of other roles first.

## 5 Implementation and Experiments

We implemented the individual derivations cache with the sketched extensions in the tableau-based reasoning system Konclude with minor adaptations to fit the architecture and the optimisations of Konclude. In particular, we use Konclude's efficient, but incomplete saturation procedure [15] to initialise the cache entries for all individuals. If completeness of the saturation cannot be guaranteed for an individual, we mark the corresponding cache entry as inconsistent such that it is reprocessed with the tableau algorithm. Parallel processing (via small batches) is straightforward for the saturation as individuals with their assertions are handled separately. This automatically leads to a very efficient handling of the "simple parts" of an ABox and it only remains to implement the (repeated) reprocessing of individuals with inconsistent cache entries.

Since tableau algorithms are usually quite memory intensive, scalability of the parallelisation not only depends on the CPUs but also on the memory bandwidth and access. Hence, the memory allocator must scale well and the data must be organised in a way that allows for effectively using the CPU caches (e.g., by writing the data of entries with one thread in cohesive memory areas). We investigated different memory allocators (hoard, tcmalloc, jemalloc) and integrated jemalloc [7] since it seems to work best in our scenario. The worker threads for constructing completion graphs only extract the data for cache updates. A designated thread then integrates the cache updates, based on the update ids introduced on page 8, which reduces blocking, improves memory management, and allows for more sophisticated update mechanisms.

To further improve the utilisation of multi-processor systems and to avoid bottlenecks, we also parallelised some other processing steps, e.g., parsing of large RDF triple files, some preprocessing aspects (i.e., extracting internal representations from RDF triples), and indexing of the cache entries for retrieving candidates for query answering. Also note that some higher-level reasoning tasks of Konclude are already (naively) parallelised by creating and processing several consistency checking problems in parallel.

For evaluating the approach,[1] we used the large ontologies and the appertaining queries from the PAGOdA evaluation [21], which includes the well-known LUBM and UOBM benchmarks as well as the real-world ontologies ChEMBL, Reactome, and Uniprot[2] from the European Bioinformatics Institute. To improve the evaluation w.r.t. the computation of large amounts of answers, we further include the queries from tests for the datalog engine VLog [4], but we use them w.r.t. the original TBoxes. We run the evaluations on a Dell PowerEdge R730 server with two Intel Xeon E5-2660V3 CPUs at 2.4 GHz and 512 GB RAM under a 64bit Ubuntu 18.04.3 LTS. For security reasons and due to multi user restrictions, we could, however, only utilise 480 GB RAM and 8 CPU cores of the server in a containerised environment (via LXD).

Metrics of the evaluated ontologies are depicted on the left-hand side of Table 3, whereas the right-hand side shows the (concurrent) parsing times for the ontologies in seconds, where K-1, K-2, K-4, and K-8 stand for the versions of Konclude, where 1,

---

[1] Source code, evaluation data, all results, and a Docker image (koncludeeval/parqa) are available at online, e.g., at https://zenodo.org/record/4606566.

[2] We evaluated query answering on a sample (denoted with $Uniprot_{40}$) since the full Uniprot ontology ($Uniprot_{100}$) is inconsistent and, hence, not interesting for evaluating query answering.

Table 3: Evaluated ontologies, number of queries (from the PAGOdA+VLog evaluation), and parsing times with different number of threads in seconds (speedup factor in parentheses)

| Ontology | DL | #Axioms | #Assertions | #Q | K-1 | K-2 | K-4 | K-8 |
|---|---|---|---|---|---|---|---|---|
| ChEMBL | $\mathcal{SRIQ(D)}$ | 3,171 | $255.8 \cdot 10^6$ | $6+3$ | 1830 | 935 (2.0) | 497 (3.7) | 268 (6.8) |
| LUBM$_{800}$ | $\mathcal{ALEHI^+(D)}$ | 93 | $110.5 \cdot 10^6$ | $35+3$ | 363 | 184 (2.0) | 102 (3.6) | 56 (6.5) |
| Reactome | $\mathcal{SHIN(D)}$ | 600 | $87.6 \cdot 10^6$ | $7+3$ | 66 | 34 (1.9) | 19 (3.6) | 11 (6.2) |
| Uniprot$_{100}$ | $\mathcal{ALCHOIQ(D)}$ | 608 | $109.5 \cdot 10^6$ | $-^2$ | 409 | 229 (1.8) | 119 (3.5) | 63 (6.7) |
| Uniprot$_{40}$ | $\mathcal{ALCHOIQ(D)}$ | 608 | $42.8 \cdot 10^6$ | $13+3$ | 215 | 113 (1.9) | 59 (3.6) | 33 (6.5) |
| UOBM$_{500}$ | $\mathcal{SHIN(D)}$ | 246 | $127.4 \cdot 10^6$ | $20+0$ | 431 | 227 (1.9) | 121 (3.5) | 66 (6.5) |

2, 4, and 8 threads are used, respectively. Since the parallel parsing hardly requires any synchronisation and only accesses the memory in a very restricted way, it can be seen as a baseline for the achievable scalability (there are minor differences based on how often the different types of assertions occur).

The left-hand side of Table 4 shows the (concurrent) pre-computation times, i.e., the time that is required to get ready for query answering after parsing the ontology, which includes the creation of the internal representation, preprocessing the ontology (e.g., absorption), saturating the concepts and individuals, repeatedly reprocessing the individuals with inconsistent cache entries, classifying the ontology, and preparing data structures for an on-demand/lazy realization. Consistency checking clearly dominates the (pre-)computation time such that the other steps can mostly be neglected for the evaluation (e.g., classification takes only a few milliseconds for these ontologies). As shown in Table 4, our parallelisation approach with the individual derivations cache is able to significantly reduce the time required for consistency checking, but the scalability w.r.t. the number of threads depends on the ontology. For LUBM and ChEMBL, the approach scales almost as well as the parsing process, whereas the scalability w.r.t. Reactome seems limited. The Reactome ontology intensively relies on (inverse) functional roles such that many and large clusters of same individuals are derived in the reasoning process. With a naive implementation of the cache, we would store, for each individual in a cluster, all derived neighbour relations, which easily becomes infeasible if large clusters of same individuals are linked. For our implementation of the cache, we identify and utilise representative individuals to store the neighbour relations more effectively, but we require consistent cache entries for this. If the clusters of same individuals are updated in parallel, which often leads to inconsistent cache entries, more neighbour relations must be managed and, thus, the parallelisation of ontologies such as Reactome only works to a limited extent with the current implementation. Also note that the enormous amounts of individuals in these ontologies make it impossible for the previous version of Konclude to build full completion graphs covering the entire ABox, i.e., the version of Konclude without the cache quickly runs out of memory for these ontologies. Also note that the individuals from the cache are mostly picked in the order in which they are indexed, i.e., more or less randomly due to hashing of pointers. Nominals, however, are indexed first and, hence, are prioritised in the (re-)processing. Clearly, the processing order can have a significant influence on how much (re-)processing is re-

Table 4: (Pre-)computation and accumulated query answering times for the evaluated ontologies with different numbers of threads in seconds (speedup factor in parentheses)

| Ontology | (Pre-)computing | | | | Query answering | | | |
|---|---|---|---|---|---|---|---|---|
| | K-1 | K-2 | K-4 | K-8 | K-1 | K-2 | K-4 | K-8 |
| ChEMBL | 2421 | 1244 (1.9) | 663 (3.7) | 397 (6.1) | 12767 | 8927 (1.4) | 4507 (2.8) | 3231 (4.0) |
| LUBM$_{800}$ | 2793 | 1658 (1.7) | 831 (3.4) | 437 (6.4) | 2777 | 1829 (1.5) | 1026 (2.7) | 569 (4.8) |
| Reactome | 1408 | 687 (2.0) | 427 (3.3) | 361 (3.9) | 935 | 524 (1.8) | 333 (2.8) | 232 (4.0) |
| Uniprot$_{100}$ | 1343 | 742 (1.8) | 429 (3.1) | 302 (4.4) | N/A[2] | N/A[2] | N/A[2] | N/A[2] |
| Uniprot$_{40}$ | 1090 | 532 (2.0) | 289 (3.7) | 198 (5.5) | 28 | 21 (1.3) | 16 (1.8) | 14 (2.0) |
| UOBM$_{500}$ | 1317 | 735 (1.8) | 394 (3.3) | 245 (5.4) | 3774 | 1799 (2.1) | 947 (4.0) | 554 (6.8) |

quired, but the runs for the evaluated real-world ontologies showed hardly any variance since most consequences could be derived locally.

The right-hand side of Table 4 reveals the query answering times (and scalability), accumulated for each ontology. Since not all steps are parallelised and the version of Konclude with only one thread uses specialised and more efficient implementations in some cases (e.g., an optimised join algorithm for results from several sub-queries, whereas the parallelised version is based on several in-memory map-reduce steps), query answering scalability leaves still room for improvement. Nevertheless, without splitting the propagation tasks, several queries cannot be computed, i.e., the version of Konclude without the presented (query answering) splitting techniques cannot answer all of the queries within the given memory and time limits. Moreover, the parallelisation significantly improves the query answering times and the improvements are larger, the more computation is required. As a comparison, PAGOdA requires 19, 666 s for loading and preprocessing all ontologies and more than 101, 817 s for query answering, where it reached the memory limit for one query and for two the time limit of 10 hours.

## 6   Conclusions

We show how the now ubiquitous multi-core processors can be used for parallelising reasoning tasks such as consistency checking and (conjunctive) query answering for expressive Description Logics. For this, we split the assertional knowledge of an ontology to similarly sized work packages for tableau-based reasoning. The technical foundation is a cache that stores chosen consequences derived for individuals and appropriate expansion as well as cache maintenance strategies to ensure correctness and termination. Our experiments with the reasoning system Konclude show promising performance improvements. The approach may even be a suitable basis for distributed reasoning in a compute cluster, where cache entries are distributed over different machines, and for incremental/stream reasoning, where a few assertions are (frequently) added or removed.

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
