# OpenReview forum: "Parallelised ABox Reasoning and Query Answering with Expressive Description Logics"
_eswc-conferences.org/ESWC/2021/Conference/Research_Track — ESWC 2021 Research_

### Official Review · AnonReviewer3 · 2021-01-14
**Good paper**

**Rating:** 2
**Confidence:** 4
**Impact:** 3
**Design And Technical Quality:** 3

**Review:**

The paper provides a parallel approach to build completion trees to improve query answering computation time.
Overall, the paper is well written. However, the paper can be followed likely by a Description Logics adept only, but that's a consequence of the space limitations. An experimental evaluation (data available, except ABox) has been conducted that shows the improvements that one may obtain by means of the method.

After rebuttal: I acknowledge to have read the authors' rebutal

**Anonymity:**

No, I would like my review to be deanonymized.

**Reuse And Availability:**

4: High

**Strong Points:**

- non negligible query answering time improvement for large size ABoxes

**Subreviewer:**

I submitted this review.

**Weak Points:**

- the paper can be followed likely by a Description Logics adept only, but that's a consequence of the space limitations

---

> ### Author Rebuttal · Authors · 2021-01-29
>
> We thank the reviewer for the invested time and the suggestions for improving the paper.
>
> The ABoxes are indeed not contained in the provided zip file, but there is a script that automatically downloads them from the PAGOdA evaluation files and, to the best of our knowledge, the downloads still work. We will probably create a zenodo record with all data included for the final version of the paper, but we did not want to create and upload too many temporary versions since these ABox files are quite big (even when zipped).

---

### Official Review · AnonReviewer4 · 2021-01-14
**A paper with good contents, but its results and methods should be better focused**

**Rating:** 1
**Confidence:** 3
**Impact:** 3
**Design And Technical Quality:** 4

**Review:**

The paper presents a method for the parallelization of query answering over large ABoxes, based on the definition of a suitable caching mechanism for the synchronization of the reasoning over the different ABox partitions.

The paper first introduces the problem of reasoning with large assertional knowledge in DL knowledge bases and summarizes the current approaches. After a brief introduction to DLs and tableaux methods, the authors introduce their definition for parallelization of ABox reasoning via the definition of Individual Cache and its use in the processing of ABox partitions. The parallelization approach is then extended to Query Answering by combining it with the absorption approach recently introduced in [13].
Finally the parallelization method is evaluated with respect to known DL ontology datasets, by assessing reasoning on Konclude with increasing parallel threads.

In general, the caching method provides a quite natural solution for the synchronization of parallel reasoning procedures: the experimental results highlight its viability for reasoning over large assertional knowledge bases.
The technical quality of the work appears to be solid and the experimental results are supported by the availability of the framework and evaluation data.

On the other hand, the paper misses a clear formal statement on the formal results of the proposed methods (in particular completeness and termination) and to which expressive DL languages it can be applied (some result is reported in the accompanying technical report).
In general, the properties of the approach are mostly discussed at a high level, while a more formal statement (and possibly an exemplification on the running example) would help to understand the advantages of the approach.
In particular, the effect of balancing the "weight" of the different ABox partitions on  reasoning problems (and the presented experiments) should be clarified.

There is no clear discussion of the related works and how this approach can surpass what it is possible with other similar methods (e.g. the approaches cited in the introduction).
This discussion can also help understanding the novelty and significance of the work with respect to the current methods.

Since some of the contributions of the work are presented in the external technical report, it might be good to re-organize the paper to move some of the discussion externally and clarify the above aspects.


**Anonymity:**

Yes, I would like my review to remain anonymous.

**Reuse And Availability:**

4: High

**Strong Points:**

- The paper provides a principled method for the parallelization of reasoning over large DL knowledge bases.
- The experimental results appear to support the viability of the method over well-known ontology datasets.
- The evaluation framework and data is available for replication.

**Subreviewer:**

I submitted this review.

**Weak Points:**

- Formal results establishing the correctness and termination of the method are not included in the main paper and the applicability of the approach to different DL languages is not clearly discussed.
- Presentation of the method and its benefits (w.r.t. current approaches for dealing with large ABoxes) should be enhanced (possibly by further examples).

---

> ### Author Rebuttal · Authors · 2021-01-29
>
> We thank the reviewer for the invested time and the suggestions for improving the paper.
>
> The proposed caching technique is designed for the Description Logic SROIQ and we also implemented it in Konclude for handling SROIQ/OWL 2 DL ontologies. The referred query answering technique also works for SROIQ as long as only a limited number of new nominals is enforced (as discussed in the paper), but this is generally the case in practice.
>
> Due to the space limitations, it seems rather difficult to integrate all relevant theoretical and practical results in the actual paper. Since parallelisation has a strong practical aspect, we focused more on the practical than the theoretical aspects in the submission and left other details for the technical report (it is more or less straightforward to adapt the compatibility criteria such that all cases for SROIQ are covered). In practice, one may also adapt the criteria if it better suits the (optimisations in the) reasoning system, i.e., there is quite some flexibility on how to realise the compatibility criteria and what to store in the cache. But we will have a critical look at the paper and we will try to highlight some theoretical aspects.
>
> We also agree that the discussion with related work could be more thorough. As indicated in the introduction, existing approaches cannot handle conjunctive queries with arbitrary existential variables (since these queries require fully-fledged reasoning procedures such as tableau algorithms and because it is unclear how to split/partition the data for such queries upfront). Moreover, most methods can only deal with fragments of SROIQ.

---

### Official Review · AnonReviewer1 · 2021-01-15
**A parallel optimization for reasoning with expressive ontologies having large aboxes**

**Rating:** 2
**Confidence:** 3
**Impact:** 3
**Design And Technical Quality:** 3

**Review:**

This paper presents a novel method to efficiently process aboxes in parallel.
The algorithm for caching partial results and the query answering addition are well described.
Tests with Konclude seem to reveal a significant potential for future benefit and investigation.

(was unable to run the zipped evaluation code on my computer, otherwise would have written a longer review)

**Anonymity:**

Yes, I would like my review to remain anonymous.

**Reuse And Availability:**

3: Medium

**Strong Points:**

Well written

Thorough explanation of caching procedure and queries

Good potential improvement to reasoning speeds for high expressivity ontologies with aboxes

**Subreviewer:**

I submitted this review.

**Weak Points:**

Algorithm descriptions and mathematical definitions in the paper tend to be a bit dense and technical (of course). However, it may occasionally be easier to provide source and discuss at a higher level

---

> ### Author Rebuttal · Authors · 2021-01-29
>
> We thank the reviewer for the invested time and the suggestions for improving the paper.
>
> The evaluation should run on all reasonably up-to-date Linux systems, but some ontologies/queries indeed require a significant amount of main memory. The evaluation package further contains the source code of the evaluated version of Konclude, but compiling it for other platforms is non-trivial due to the Redland RDF Library dependency. We will also provide a docker image for the final version of the paper to further facilitate the reproduction of the results.

---

### Official Review · AnonReviewer5 · 2021-01-15
**Solid paper, well written**

**Rating:** 2
**Confidence:** 4
**Impact:** 3
**Design And Technical Quality:** 4

**Review:**

Summary

Authors describe a procedure to split the ABox axioms and parallelize the reasoning (consistency checking) and conjunctive query answering operations. Tableau algorithm is used to construct the completion graph of the equally sized splits of ABox axioms. A cache specifically designed for this approach is used for controlling and synchronizing the interactions between the constructed partial models of the different splits. For conjunctive query answering, authors use their earlier work of absorbing query atoms into axioms and use completion graph. The proposed parallel algorithm has been implemented in the Konclude reasoner and the evaluation results show good speedup.

Review

Parallelizing tableau algorithms that work in practice is challenging. So from that point of view, it is quite pleasing to see this work and the results. This work could pave the way for further research in this direction. Along with parallelizing the consistency check reasoning task, the authors also parallelize conjunctive query answering, which is generally ignored by other related work. Having said that, the major parts of the most important sections of this paper (Sections 3 and 4) are quite dense and is a hard read. Introducing the example early and explaining each step of the algorithm would help in improving the readability. Some of the textual descriptions of the algorithm could be removed to make space.

In the evaluation, comparison with other reasoners, especially some of the parallel and efficient reasoners such as ELK (for the EL ontologies), RDFox and VLog is missing. It would be good to have this comparison to get a perspective on the efficiency of the parallelized Konclude. Would it be possible to include at least a brief comparison in the current submission?

Other questions/comments
  - In Table 4, it would be good to know the point (number of threads) at which the curve flattens and comes down (performance drops).
  - How generalizable is this approach on the tableau algorithms of other reasoners such as Pellet and HermiT?
  - Is locking involved? If not, how is it bypassed?

Post rebuttal: Thank you for answering my questions. Please add the points from the rebuttal in the paper.

**Anonymity:**

Yes, I would like my review to remain anonymous.

**Reuse And Availability:**

2: Low

**Strong Points:**

  - Parallelizing tableau algorithms is not easy, which makes this a very good contribution.
  - The technical quality of the paper is high and the paper is generally well-written (except for the few dense portions).


**Subreviewer:**

I delegated this review to a subreviewer.

**Weak Points:**

  - Parts of the paper are dense, which makes it hard to read.
  - Comparison with other (parallel) reasoners is missing.

---

> ### Author Rebuttal · Authors · 2021-01-29
>
> We thank the reviewer for the invested time and the suggestions for improving the paper.
>
> A comparison with other parallelised reasoning systems seems difficult. As far as we know, the parallelisation capabilities of VLog are rather limited. RDFox is a proprietary system, which makes getting a version/license and publishing results difficult. However, PAGOdA uses RDFox as underlying datalog engine (by reducing ontologies to lower and upper bound approximations), but it does not seem to (effectively) utilise the parallelisation capabilities of RDFox for consistency checking and query answering. Moreover, these other parallelised reasoners focus on much simpler language fragments such that a direct comparison is probably not very helpful (reducing the expressiveness of the ontologies to the jointly supported features would result in trivial TBoxes). To have a baseline for the achievable scalability, we added the times/speedup factors for parallelised parsing.
>
> We did some limited experiments with a higher number of threads (e.g., 16) and for most of the evaluated ontologies, there are still some but diminishing performance gains. (Parallelisation of Reactome gets more and more difficult due to large clusters of individuals.) At the moment, one thread manages the updates for the cache and at some point it will become necessary to also partition the cache (which should be possible, but requires some work).
>
> In principle, the approach can be implement also in other tableau-based reasoners, but to maximise efficiency some adaptation might be needed. For example, the realisation procedure should directly access the cache for determining (possible) instances. An implementation in HermiT seems, however, difficult due to the differences between tableau and hyper-tableau algorithms (the latter use a semi-naive evaluation of clauses instead of the typical application of tableau expansion rules).
>
> Our implementation works without locking the cache by sending update messages to a thread that manages the cache. Concurrent read access is ensured by appropriately designed data structures and an update mechanism that temporarily works on copies (of parts) of entries.

---

### Official Review · AnonReviewer2 · 2021-01-17
**Good practical approach with some missing formal justifications**

**Rating:** 2
**Confidence:** 4
**Impact:** 3
**Design And Technical Quality:** 4

**Review:**

  The paper proposes a practical approach for reasoning with ontologies
  containing large number of facts about individuals. More precisesly,
  it presents a method for parallelising ABox reasoning an query answering
  for knowledge bases with large number of assertions. The approach is
  a rather practical one based on using a cache that stores and maintains
  information about individuals. These informations are generated during
  the reasoning process.

  There have been previously similar approaches to parallelise either the
  TBox reasoning or the ABox reasoning, so the idea is not brand new. But
  the technical details of the apprach, based on using a cache differ from the
  previous ones. The paper is understandable, the approach is described
  throughout the text and is accompanied by a set of tableau rule extensions
  (Table 2) and an algorithm (Algorithm 1).

  My main critisim about the paper is the lack of formal justification for
  some of the arguments in the paper. For instance an analysis of Algorithm
  1 is missing. Similarly the correctness and completeness of the newly
  introduced rules in Table 2 are not discussed.

  The experimental results suggest that the approach indeed speeds up the
  reasoning in practice, this is nice to see. The computer used for the
  experiments has a huge main memory, which is not often the case on a
  usual desktop computer. Is this amount of memory necessary, what is the
  minimum amount of memory required for the approach, can I have this method
  run on my desktop computer? It would have been good to report about these
  as well.

  Table 2 and algorithm 1: separate them for better readability


**Anonymity:**

Yes, I would like my review to remain anonymous.

**Reuse And Availability:**

3: Medium

**Strong Points:**

practical approach

**Subreviewer:**

I submitted this review.

**Weak Points:**

missing formal justification

---

> ### Author Rebuttal · Authors · 2021-01-29
>
> We thank the reviewer for the invested time and the suggestions for improving the paper.
>
> Due to the space limitations, we left the formal proof of completeness for the caching technique as well as explanations why soundness and termination (trivially) hold for the technical report. Note that the rules of Table 2 are not complete (blocking is omitted for the sake of brevity) and they mainly recapitulate an already published approach for query answering. We will clarify that correctness/soundness aspects regarding this approach can be found in the corresponding paper.
>
> We agree that there is no thorough analysis of Algorithm 1, which main purpose is, however, to give an idea how the splitting can be realised. In fact, we implemented it slightly differently in the actual reasoning system since a recursive algorithm is not very suitable for creating several propagation tasks in parallel (the recursion means that one has to wait until the completion graph construction process is finished before creating the next one), which is discussed in the technical report in more detail and we will try to (briefly) address this in the paper, too.
>
> The required main memory heavily depends on (the size of) the ontologies. Several aspects of Konclude are not yet designed to be very memory efficient. For example, Konclude keeps the parsed axioms/triples in the main memory to facilitate incremental updates, which, however, also requires a significant amount of memory (up to 100GB for the evaluated ontologies). Some of the queries (e.g., ChEMBL) result in billions of answers, which are computed and returned in a stream-based manner, but Konclude still needs almost all available memory for their computation. To sum up, these extremely large ontologies can, at the moment, not be run on a usual desktop computer, but the caching techniques significantly reduce the memory consumption (without the caching techniques, Konclude even runs quickly out of memory on the server for these ontologies).

---

### Decision · Program_Chairs · 2021-02-23

**Decision:**

Accept

**Comment:**

All the reviewers agree that the paper should be accepted. One limitation of the current paper is that it does not provide formal justifications about correctness/completion etc., which is most likely due to space restrictions. However, some of these arguments are provided in another technical report. We encourage the authors to publish the technical report in a persistent archive (e.g., arxiv), and to link it to the article so that the interested reader can find more information there.